# Probenecid Inhibits Respiratory Syncytial Virus (RSV) Replication

**DOI:** 10.3390/v14050912

**Published:** 2022-04-27

**Authors:** Jackelyn Murray, Harrison C. Bergeron, Les P. Jones, Zachary Beau Reener, David E. Martin, Fred D. Sancilio, Ralph A. Tripp

**Affiliations:** 1Department of Infectious Diseases, University of Georgia, Athens, GA 30602, USA; jcrab@uga.edu (J.M.); harrison.bergeron@uga.edu (H.C.B.); lj66@uga.edu (L.P.J.); zbr17323@uga.edu (Z.B.R.); 2TrippBio, Inc., Jacksonville, FL 32256, USA; davidmartin@trippbio.com; 3Department of Chemistry and Biochemistry, Florida Atlantic University, Jupiter, FL 33431, USA; fredsancilio@clearwayglobal.com

**Keywords:** respiratory syncytial virus (RSV), antiviral, prophylactic, therapeutic

## Abstract

RNA viruses like SARS-CoV-2, influenza virus, and respiratory syncytial virus (RSV) are dependent on host genes for replication. We investigated if probenecid, an FDA-approved and safe urate-lowering drug that inhibits organic anion transporters (OATs) has prophylactic or therapeutic efficacy to inhibit RSV replication in three epithelial cell lines used in RSV studies, i.e., Vero E6 cells, HEp-2 cells, and in primary normal human bronchoepithelial (NHBE) cells, and in BALB/c mice. The studies showed that nanomolar concentrations of all probenecid regimens prevent RSV strain A and B replication in vitro and RSV strain A in vivo, representing a potential prophylactic and chemotherapeutic for RSV.

## 1. Introduction

Respiratory syncytial virus (RSV) is the leading viral pathogen associated with lower respiratory tract disease in infants and young children worldwide also afflicting the elderly and immune compromised [1,2]. Preventing RSV morbidity and mortality has been an effort of research and vaccine studies development for decades. RSV is responsible for >150,000 pediatric hospitalizations/year costing >$300 million in young children based on health care costs of hospitalization of young children for RSV infections [3]. Approved therapeutic intervention is limited to inhaled ribavirin and palivizumab (Synagis), a humanized monoclonal antibody targeting the F protein. Ribavirin has shown mixed-to-poor results and palivizumab treatment is not fully effective [4,5]. Additionally, palivizumab is administered monthly to help protect high-risk infants from severe RSV disease throughout the RSV season, and although treatment reduces hospitalizations in infants by approximately 50%, its efficacy decreases as mutations in F protein are induced by treatment [6,7]. Unfortunately, there is no safe and effective RSV vaccine available despite years of effort, thus there is a need for effective RSV therapeutics.

As an alternative to developing novel antiviral drugs, repurposed or repositioned drugs with known safety profiles could reduce costs and the time needed for the development of new drugs. One example is the drug minocycline that is a tetracycline antibiotic with efficacy in bacterial infections as well as antiviral activity against influenza virus and other viruses including RSV in vitro [8]. Improvement of the anti-RSV drug repertoire could be addressed by having a greater understanding of virus-host interactions as this pathway has led to the discovery and validation of new therapeutic drug targets for disease intervention [9]. Antiviral drugs could target either viral proteins or cellular proteins. Targeting the virus with antiviral drugs increases the likelihood of developing drug resistance. In contrast, a drug that targets a host cell protein used for virus replication could inhibit multiple viruses using that pathway.

By combining high-throughput screening (HTS) with RNA interference (RNAi), host gene silencing can lead to the rapid discovery of host genes and pathways for developing antiviral treatments [10,11,12,13]. We previously used RNAi to discover and validate drug targets as a means to filter and prioritize therapeutics [13,14,15]. Specifically, we used genome-wide RNAi screens to distinguish pro- and anti-viral host genes that affect virus replication that resulted in repurposed drugs to inhibit influenza A virus (IAV) replication [13,16]. Using an RNAi screen to determine host genes from respiratory epithelial (A549) cells on influenza A virus replication (IAV), we discovered that organic anion transporter-3 (OAT3) is required for IAV replication [13]. We showed that probenecid treatment inhibited the OAT3 (SLC22A8) gene and reduced IAV replication in vitro and in mice [13]. The SLC family of solute carriers is expressed in both human and mouse tissues [17], and transfection of A549 cells with siRNA targeting the OAT3 gene completely silenced IAV replication. We showed that probenecid dramatically reduced IAV replication in vitro (IC_50_ = 5 × 10^−5^ to 5 × 10^−4^ uM; *p* < 0.05), and mice treated daily over 3 days with 25 mg/kg probenecid following lethal challenge (2 × 10^3^ PFU/mouse) with mouse-adapted IAV (A/WSN/33) were partially protected (60% survival; *p* < 0.05). We also showed that RNAi silencing of closely related transporters, i.e., OAT1, OAT2, OAT4, OAT7, and URAT1 did not affect IAV replication indicating a specific role of OAT3 to support IAV replication [13]. The OAT3 gene has 12 predicted transmembrane domains, is principally expressed in the kidney [18], and is important for urinary excretion of anionic metabolites. No gender-based differences in OAT3 expression have been reported in humans [19]. Although much focus has been on the kidney, OATs are localized to almost all epithelial barriers in the body [17,20]. OAT3 is expressed in both human and mouse lung respiratory epithelial cells. Probenecid is a uricosuric agent, a chemical inhibitor of OAT3, and a well-described treatment for gout, and is a favorable candidate for antiviral drug repurposing because it is commercially available and has a benign clinical safety profile [21]. Probenecid also affects other ion channels and may affect inflammatory responses [22].

Viruses are dependent on co-opted host genes for replication [23]. To determine if probenecid prophylaxis or treatment inhibited RSV replication, we tested nanomolar to micromolar concentrations of probenecid to prevent RSV strain A and strain B replication in epithelial cell lines and mice. Three cell lines, i.e., Vero E6 cells, HEp-2 cells, and undifferentiated primary normal human bronchoepithelial (NHBE) cells, and male and female BALB/c mice were examined. The studies show that probenecid significantly reduces RSV replication in vitro and in vivo. These results are consistent with previous findings for influenza [13] and SARS-CoV-2 [24] suggesting that probenecid regimens are likely transferrable to other respiratory viruses which utilize solute carriers during replication representing a potential host-directed pan-anti-viral.

## 2. Materials and Methods

### 2.1. Cells and Cell Culture

Vero E6 cells (ATCC; CRL-1586), a cell line clone isolated from the kidney of an African green monkey, and HEp-2 cells (ATCC; CCL-23), a cell line that was cloned from human larynx cells and determined by karyotyping to be free of HeLa cell contamination were propagated in Dulbecco’s modified Eagle’s medium (DMEM; Gibco | Thermo Fisher Scientific, Waltham, MA USA) supplemented with 5% heat-inactivated fetal bovine serum (FBS; Hyclone, Logan, UT, USA) at 37 °C with 5% CO_2_. Vero E6 cells and HEp-2 cells were maintained in log-phase in T75 cm^2^ culture flasks (ThermoFisher, Waltham, NA, USA) and HEp-2 was used for virus propagation. HEp-2 and Vero E6 cells depend largely on RSV G protein binding to cell surface glycosaminoglycans (GAGs). GAG-dependent infection is reduced by a single passage of RSV in Vero E6 cells [25]. Normal human bronchial epithelial (NHBE) cells (Lonza Bioscience, Basel, Switzerland) from a healthy male donor were expanded, cryopreserved, and maintained in bronchial epithelial cell growth medium (BEGM; Lonza) through 15 population doublings and were used undifferentiated.

### 2.2. Viruses

RSV A2 (ATCC VR-1540) and RSV B1 (ATCC VR-1580) were obtained from the American Type Culture Collection (ATCC), Manassas, Virginia, or Memphis-37 (a clinical strain of human RSV strain A) was obtained from Meridian Life Science, Memphis, TN, USA) were propagated and quantified on HEp-2 cells and Vero E6 cells then stored at −80 °C as described previously [26]. HEp-2 cells and Vero E6 cells were maintained in Dulbecco’s modified essential medium (DMEM) supplemented with glutamine and 5% fetal bovine serum (5% DMEM; Gibco). Virus titers were determined using a methylcellulose plaque assay as described [27].

### 2.3. In Vitro Probenecid Inhibition Assays

A working stock of probenecid (Sigma-Aldrich, St. Louis, MO, USA) was dissolved in DMSO (Sigma) and dilutions of the working stock were resuspended in PBS (Gibco). Cellular toxicity was determined using a ToxiLight Bioassay (Lonza). Vero E6 cells, HEp-2 cells, or undifferentiated NHBE cells were plated overnight at 10^4^ cells/well in 96-well flat-bottom plates (Costar). Cells were either pretreated for 24h prior to infection (prophylactically) or therapeutically at 24h post-infection with probenecid at different concentrations, i.e., 100, 50, 25, 12, 6, 3, 1, 0.5, 0.2, 0.1, 0.05, 0.01, or 0 µM. Subsequently, the media and probenecid were removed and the cells were infected with RSV A2, RSV B1, or Memphis-37 at MOI = 0.1. At 72 h post-infection the plates containing the cells were frozen at −80 °C the freeze-thawed 3X and the cell-free supernatants were used for log10 dilutions in RSV plaque assays.

### 2.4. In Vivo Inhibition Studies

BALB/c male and female mice (6–8 weeks old) were obtained from Charles River and rested a week before use. The animal study protocol was approved by the Institutional Review Board of the University of Georgia, A2021 03-006-Y2-A0, Immunity to Respiratory Viruses and Virus Proteins in Mus musculus, approved on: 6 May 2021. All experiments were performed with five mice per group and repeated twice independently. To evaluate lung virus titers, probenecid was administered intraperitoneally (i.p.) at doses and time points pre- or post-RSV infection as indicated in the Results. Briefly, 2 mg/kg or 200 mg/kg of probenecid in PBS were i.p. delivered to the mice. On days 3, 5, and 7 bronchoalveolar lavage (BAL) samples were collected from individual mice and analyzed. BAL cell yield was determined by counting the total cell number, and cell viability was determined by Trypan blue (Sigma) exclusion. Smears for cell differentiation were prepared by cytocentrifugation (Shandon), and cell differentiation was performed by microscopy on cytospun slides after staining with hematoxylin and eosin staining where at least 100 cells were counted for macrophages, polymorphonuclear (PMN) cells, lymphocytes, and eosinophils [28]. At each time point, sera were collected, and the lungs were isolated to determine virus titers by PFU/mL analyses [13]. For virus titration analyses, lung homogenates were serially diluted, and the titer was determined on Vero E6 cells [29].

The BAL cell pattern reflects the inflammatory cell profile in the lung [28]. Neither prophylactic nor therapeutic probenecid treatment with 2 mg/kg or 200 mg/kg probenecid had substantial effects on the differential cell counts or BAL leukocyte subpopulations at days 3, 5, or 7 pi (Appendix A). Further, no substantial differences in BAL cells were evident by smears despite the reduced RSV lung titers in the probenecid-treated mice highlighting the anti-RSV effects of the drug.

### 2.5. Lung Virus Titers

Lung viral titer from RSV-infected mice was determined as previously described [26]. Briefly, lungs were homogenized in 1 mL of sterile Dulbecco PBS per lung, and 10-fold serial dilutions in serum-free DMEM (Gibco) were added to confluent Vero cell monolayers in 24-well plates. After adsorption for 2 h at 37 °C, cell monolayers were overlaid with 2% methylcellulose, incubated at 37 °C for 6 days, and then enumerated by immunostaining with anti-F protein monoclonal antibody, 131-2A.

### 2.6. OAT3 Expression

SLC22A8 (OAT3) transcripts were evaluated by qPCR as previously described [30]. For in vitro studies, HEp-2 cells were plated in 96-well tissue culture plates (Corning Life Sciences, Durham, NC, USA) and treated with the IC_90_ of probenecid (7.2 uM) or DMSO only control for 24 h. RNA was isolated by RNAzol RT (Molecular Research Center, Cincinnati, OH, USA) and digested with DNAse1, and total RNA was quantified by Nanodrop (ThermoFisher Scientific, Waltham, MA, USA). cDNA first-strand synthesis was performed using LunaScript (New England Biolabs, Ipswich, MA, USA) as described by the manufacturer. cDNA was used as a template for qPCR in Luna Universal qPCR master mix (New England Biolabs). For in vivo studies, BALB/c lung RNA were extracted by RNAdvance Tissue (Beckman Coulter Life Sciences, Indianapolis, IN, USA) at indicated time points and processed as described above. Primer pairs used are:
**Gene****Primer Left (5’-3’)****Primer Right (5’-3’)**
SLC22A8 (huOAT3)

TGCAAATGAATGCGAATGAGG

CGGTCGTCGCATAACACATA

ACTB (huB-actin)

CATGTACGTTGCTATCCAGGC

CTCCTTAATGTCACGCACGAT

SLC22A8 (msOAT3)

CATACTCACTCCTGCACTCATC

CCAGGGAATCTCAAAGGGAAA

ACTB (msB-actin)

CAGCCTTCCTTCTTGGGTATG
GGCATAGAGGTCTTTACGGATG

Gene expression was determined and raw Ct values or fold change (reciprocal of 2ΔΔCt) are presented normalized to housekeeping gene. Data represent mean Ct values + 95% confidence interval, or SEM, respectively, of three independent repeats.

### 2.7. Statistical Analysis

Statistical analyses were done using the Student’s *t*-test or one-way analysis of variance (ANOVA), as indicated. Data were analyzed for statistical significance using appropriate statistics where *p* < 0.05 was considered statistically significant using Prism 9 (GraphPad). Results were calculated as means ± standard errors. Values of *p* < 0.05 were considered significant.

## 3. Results

In this study, we determined if RSV replication in Vero E6 cells, HEp-2 cells, or NHBE cells infected with RSV A2, RSV B1, or Memphis-37 was affected by probenecid treatment. The different epithelial cell types were pretreated (prophylaxis) with differing probenecid concentrations (i.e., 100, 50, 25, 12, 6, 3, 1, 0.5, 0.2, 0.1, 0.05, 0.01, or 0 µM) and the effect of treatment on replication determined at 72 h after infection by plaque assay. Probenecid prophylaxis resulted in a dose-dependent decrease in RSV A2 replication in all infected cells types with an IC_50_/IC_90_ = 0.07/0.63 uM in Vero E6 cells, 0.8/7.2 uM in HEp-2 cells, and 0.4/3.6 uM in NHBE cells (Figure 1a). Cell viability was examined and as expected no cellular toxicity was evident, similar to earlier studies [13,24]. Moreover, HEp-2 cells treated with IC_90_ probenecid resulted in undetectable levels of OAT3 transcripts (Appendix A). Probenecid treatment was very effective at inhibiting RSV A2 replication in all cells types (Figure 1d). The IC_50_/IC_90_ = 0.1/2.7 uM in Vero E6 cells, 1.2/10.8 uM in HEp-2 cells, and 0.3/2.7 uM in NHBE cells. The results for probenecid prophylaxis showed the highest IC_50_/IC_90_ activity in Vero E6 cells and NHBE cells.

As RSV groups A and B co-circulate, and both groups may cause infection during a single season [31], it was important to determine the probenecid susceptibility to RSV A and RSV B particularly as it has been shown that the two groups have evolved separately for a considerable period [32]. As for RSV A2, probenecid prophylaxis resulted in a dose-dependent decrease in RSV B1 replication in all infected cells types (Figure 1b). There was no IC_90_ for RSV B1 in the treated cell types because RSV B1 was not reduced 90% using the concentrations tested. Probenecid prophylaxis resulted in an IC_50_ = 0.85 uM in Vero E6 cells, 0.8 uM in HEp-2 cells, and 0.8 uM in NHBE cells (Figure 1b, Table 1). Probenecid treated Vero E6 cells infected with RSV B1 had an IC_50_ = 2.0 uM, HEp-2 cells = 0.9 uM, and NHBE cells = 1.2 uM (Figure 1e, Table 1). Similar to RSV A2 infected cells there was no cellular toxicity detected. The results showed that probenecid prophylaxis or treatment was more effective for RSV A2 infected cell types compared to RSV B1.

Memphis-37 is an RSV A strain isolated from a pediatric case and used in studies in human adult subjects [33]. Memphis-37 that is propagated in Vero E6 cells have been shown to develop a truncated G protein [25], thus the Memphis-37 strain used in these studies was propagated in HEp-2 cells. Probenecid prophylaxis was effective at inhibiting Memphis-37 replication in all infected cells types (Figure 1c). The IC_50_/IC_90_ = 0.03/0.27uM in Vero E6 cells, 0.04/0.36 uM in HEp-2 cells, and 0.16/1.44 uM in NHBE cells (Table 1), and no effect on cell viability was detectable for any probenecid concentration. Treatment with probenecid inhibited Memphis-37 replication in all infected cells types as expected and was similar to RSV A2 and B1 studies (Figure 1f). The IC_50_/IC_90_ = 0.4/3.6 uM in Vero E6 cells, 0.5/4.5 uM in HEp-2 cells, and 0.2/1.8 uM in NHBE cells (Table 1).

Having shown probenecid to have potent activity on prophylactically or therapeutically treated cell types (Figure 1a–f); we determined the effectiveness of prophylactic or therapeutic treatment in a BALB/c mouse model of RSV infection. Male or female 6–8-week-old BALB/c mice were intranasally (i.n.) infected with RSV strain A2. Mice were treated once with probenecid 24 h before infection (prophylaxis) or 24 h post-infection (treatment) dosed at 2 mg/kg or 200 mg/kg, or with PBS. As expected, there were no substantial clinical signs of disease determined by BAL cell infiltrates (Appendix A) [34]. All probenecid regimens had significantly (*p* < 0.0001) reduced lung virus titer on days 3, 5, and 7 pi in female and male mice (Figure 2 and Figure 3, respectively). As predicted from the in vitro results), there was a considerable reduction in the lung virus load in 2 mg/kg and 200 mg/kg probenecid-treated mice challenged with RSV A2. Maximum reductions of lung virus load occurred in mice pretreated with 200 mg/kg probenecid 24 h before infection although substantial reductions in lung virus titer occurred following 2 mg/kg probenecid prophylaxis (Figure 2 and Figure 3). Mice therapeutically treated once with 2 or 200 mg/kg probenecid 24h after RSV infection also had greatly reduced RSV A2 lung titers on days 3, 5, and 7 pi (Figure 2 and Figure 3). Maximum reductions of lung virus load occurred in mice treated with 200 mg/kg probenecid, although significant (*p* < 0.0001) and substantial reductions in lung virus titers were observed in 2 mg/kg probenecid-treated mice. Moreover, RNA extracted from the lung of mice treated with 200 mg/kg probenecid had markedly reduced OAT3 transcripts compared to PBS controls 2 dpi (Appendix A).

As previously reported [24], a population pharmacokinetics (pop-PK) model was developed to characterize probenecid PK using a one-compartment structure with saturable elimination and first-order absorption. Simulations using the final pop-PK model to generate probenecid exposure profiles comparing 600 mg twice daily, 900 mg twice daily, or 1800 mg once daily administration were completed and free drug concentrations were calculated (Table 2).

## 4. Discussion

There are currently only two FDA-approved drugs for RSV: palivizumab, a monoclonal antibody for the prevention of RSV in high-risk children, and ribavirin, approved for the treatment of severe RSV disease. Both of these drugs have questionable effectiveness [35]. Despite the availability of these approved drugs, RSV remains a worldwide health concern due to the lack of a safe and effective vaccine, and limited antivirals. Recent findings from promising antiviral drug candidates [35,36] suggest improvements in RSV disease intervention [37,38]. However, drug repurposing (or repositioning) can be a method to facilitate antiviral development. Drug repurposing may focus on direct-acting antivirals or host-targeted antivirals. The majority of approved antivirals are direct-acting such as inhibitors of RNA-dependent RNA polymerase (e.g., Remdesivir) or proteases (e.g., Lopinavir). We chose to examine a host-targeted repurposed drug, i.e., probenecid, as previous siRNA studies showed that the OAT3 gene was needed for IAV replication, probenecid treatment inhibited IAV replication by reducing OAT3 [13]. Host gene pathway analysis showed OATs, which mediated transport of sodium and chloride ions across the airway lumen [39,40] are needed for viral replication. Probenecid is a chemical inhibitor of OAT transport, particularly OAT1 and OAT3 [17,41,42].

In HEp-2 cells and in BALB/c mice treated with probenecid, OAT3 expression was reduced (Appendix A), suggesting a role for OATs in RSV replication. Further, probenecid pretreatment of Vero E6 cells, HEp-2 cells, or NHBE cells was very effective at preventing RSV replication. The IC_50_ and IC_90_ of probenecid prophylaxis against RSV A2 was IC_50_/IC_90_ = 0.07/0.63 uM in Vero E6 cells, 0.8/7.2 uM in HEp-2 cells, and 0.4/3.6 uM in NHBE cells. Similarly, the IC_50_ of probenecid treatment of RSV B1 infected Vero E6 cells was IC_50_ = 0.85 uM, 0.8 uM for HEp-2 cells, and 0.8 uM for NHBE cells. Importantly, comparable IC_50_/IC_90_ results following probenecid prophylaxis or treatment of Memphis-37 infected cells were evident. These results along with the previous IAV and SARS-CoV-2 results show that nanomolar concentrations of probenecid reduce virus replication. Notably, the probenecid concentration needed for inhibition of RSV replication was substantially less than what was needed to inhibit IAV replication [13]. Administration of probenecid before (prophylaxis) or after (treatment) RSV infection reduced lung virus titers demonstrating its versatility as a chemotherapeutic. In considering the translation of these preclinical findings, it is important to consider that human plasma concentrations for probenecid are projected to exceed the protein binding adjusted IC_50_/IC_90_ value over the dosing interval providing adequate coverage against the tested strains (24). Additionally, probenecid has been tested in children aged 2 to 14 years for use together with antibiotics, and it did not cause side effects or problems. Studies on the effects of probenecid in patients with gout have only been done only in adults as gout is very rare in children. Future studies will evaluate the in vivo efficacy of probenecid against other RSV strains, determine optimal treatment regimens and dosing, as well as address probenecid’s effect on pannexin 1 channels’ important role in intercellular communication and potentially inflammation [43].

## Figures and Tables

**Figure 1 viruses-14-00912-f001:**
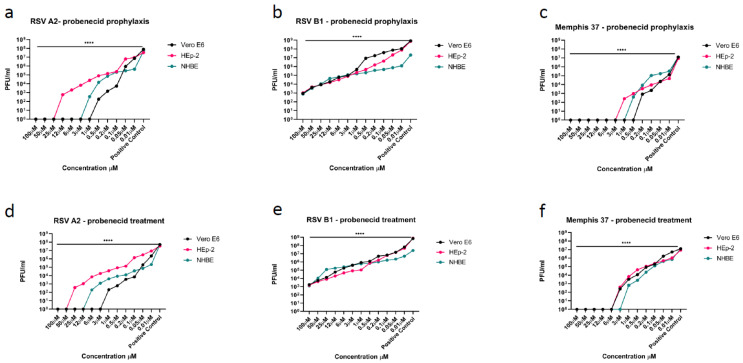
(**a**–**c**), Cell lines were prophylactically treated with probenecid 24 h prior to infection of RSV A2, RSV B1, or Memphis-37. Probenecid prophylaxis significantly (**** *p* < 0.0001) inhibited the virus replication in Vero E6 cells, HEp-2 cells, and NHBE cells compared to control (DMSO only). Viral titers were determined by plaque assay which has a limit of detection of 1 × 10² PFU/mL. The IC_50_ and IC_90_ values are shown in Table 1. (**d**–**f**), Cell lines were treated with probenecid 24h after infection of RSV A2, RSV B1, or Memphis-37. Treatment significantly (**** *p* < 0.0001) inhibited the virus replication in Vero E6 cells, HEp-2 cells, and NHBE cells compared to control (DMSO only). Viral titers were determined by plaque assay. The IC_50_ and IC_90_ values are shown in Table 1. For RSV B1 the IC_90_ values are not available as the virus was not reduced by 90%.

**Figure 2 viruses-14-00912-f002:**
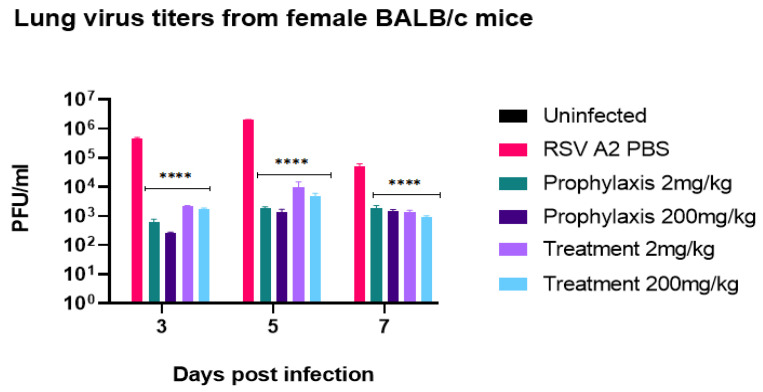
Lung virus titers from female BALB/c mice. The mice received 2 mg/kg or 200 mg/kg probenecid 24 h before infection (prophylaxis) or 24 h pi (treatment). The mice were i.n. infected with 10^6^ PFU of RSV A2. On days 3, 5, and 7 pi, the lungs were harvested, and virus titers were determined by plaque assay having a limit of detection of 1 × 10² PFU/mL. There is significant (**** *p* < 0.0001) reduction in lung viral titers with all probenecid treatments compared to control (PBS).

**Figure 3 viruses-14-00912-f003:**
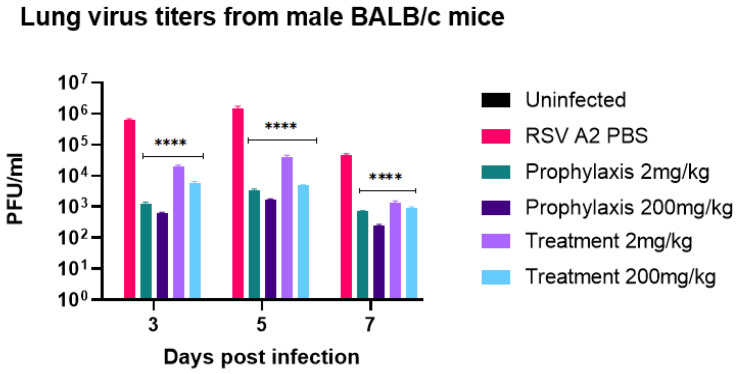
Lung virus titers from male BALB/c mice. The mice received 2 mg/kg or 200 mg/kg probenecid 24 h before infection (prophylaxis) or 24 h pi (treatment). The mice were i.n. infected with 10^6^ PFU of RSV A2. On days 3, 5, and 7 pi, the lungs were harvested, and virus titers were determined by plaque assay having a limit of detection of 1 × 10² PFU/mL. There is significant (**** *p* < 0.0001) reduction in lung viral titers with all probenecid treatments compared to control (PBS).

**Table 1 viruses-14-00912-t001:** IC_50_/IC_90_ values.

	IC_50_				IC_90_		
	RSV A2		RSV B1	Memphis-37	RSV A2	RSV B1	Memphis-37
Prophylaxis	Vero E6 cells	0.07 µM	0.85 µM	0.03 µM	0.63 µM	*	0.27 µM
	HEp-2 cells	0.8 µM	0.8 µM	0.04 µM	7.2 µM	*	0.36 µM
	NHBE cells	0.4 µM	0.8 µM	0.16 µM	3.6 µM	*	1.44 µM
Treatment	Vero E6 cells	0.1 µM	2.0 µM	0.4 µM	2.7 µM	*	3.6 µM
	HEp-2 cells	1.2 µM	0.9 µM	0.5 µM	10.8 µM	*	4.5 µM
	NHBE cells	0.3 µM	1.2 µM	0.2 µM	2.7 µM	*	1.8 µM

IC_50_ and IC_90_ values in NHBE cells, Vero E6 cells, and HEp-2 cells after treating with different probenecid concentrations and infecting with RSV A2, RSV B1, or Memphis-37. * = no IC_90_ value for the RSV B1 virus as there was not a 90% reduction of virus titers with the concentrations of probenecid used.

**Table 2 viruses-14-00912-t002:** Probenecid steady-state concentration and free drug concentrations after different probenecid doses in humans.

Dose (mg)	Frequency	Steady State Concentration (mg/mL)	Steady State Concentration (mM) with 95% Protein Binding
600	BID	30.1	5.27
900	BID	92.5	16.2
1800	QD	64.9	11.4

The doses examined are predicted to provide plasma concentrations exceeding the protein binding adjusted IC_50_/IC_90_ values for all RSV strains under all study conditions. The projected doses are below the maximum allowable FDA-approved dose and have been generally safe and well-tolerated with no significant side effects. BID = two times a day; QD = once a day.

## Data Availability

Not applicable.

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
