# Peer review of "Probenecid Inhibits Respiratory Syncytial Virus (RSV) Replication"

_viruses, 2022, doi:10.3390/v14050912_

Round 1

Reviewer 1 Report

Murray et al describe the use of the small molecule drug probenecid to inhibit replication of RSV, both in cell lines and in lungs of intranasally infected BALB/c mice.

  1. Please show the limits of detection of the plaque assays in figures 1 and 2 and adjust data points accordingly.
  2. While viral replication is certainly inhibited in lungs at the days sampled, there appears to bel a significant amount of virus present in all infected groups at day 7. Please comment on whether longer time points were investigated—in particular, is it possible that replication might increase once the drug is fully metabolized

Minor comments:

  1. In Supplementary Table 1, please show Ct values for OAT3 and Actin in the mice. Ideally, they would be paired for individual mice. The ddCt fold change can remain in addition—both are helpful.
  2. The authors mention that probenecid has a benign safety profile for use in treatment of gout. The profile of those who would benefit from probenecid as an RSV prophylactic or treatment is likely very different. Could the authors discuss any existing data on its use in young children, or potential side effects from its use.
  3. Please clarify whether the pharmacokinetics table (Table 2) pertains to humans, and if there are data to link mouse dosage to expected human effective dosage for probenecid.

Author Response

Reviewer 1: Comments and Suggestions for Authors

Murray et al describe the use of the small molecule drug probenecid to inhibit replication of RSV, both in cell lines and in lungs of intranasally infected BALB/c mice.

  1. Please show the limits of detection of the plaque assays in figures 1 and 2 and adjust data points accordingly. Jackelyn please note the LOD in the figure legends.
  • We now note in the figure legends that the viral titers were determined by plaque assay having a limit of detection of 1 X 10² PFU/ml. This has been added this to Figures 1,2 and 3.

  1. While viral replication is certainly inhibited in lungs at the days sampled, there appears to bel a significant amount of virus present in all infected groups at day 7. Please comment on whether longer time points were investigated—in particular, is it possible that replication might increase once the drug is fully metabolized.
  • We and others have previously shown that the lung virus loads in BALB/c mice infected with these and related RSV strains diminish and clear by day 7-10 pi (DOI: 10.1086/514208, doi: 10.1016/j.antiviral.2018.04.014, https://doi.org/10.1128/JVI.01693-10). Typically, peak RSV lung titers occur ~day 5 pi (e.g.106 PFU), decrease by day 7 pi (e. g.~104 PFU), and are absent by day 10 pi. Clinically, patients take probenecid over several days at a high dose to control gout. In this study, mice were treated once with probenecid, and a reduction in virus titer was observed.

Minor comments:

  1. In Supplementary Table 1, please show Ct values for OAT3 and Actin in the mice. Ideally, they would be paired for individual mice. The ddCt fold change can remain in addition—both are helpful.
  • We concur and have modified Supplementary Table 1.

  1. The authors mention that probenecid has a benign safety profile for use in treatment of gout. The profile of those who would benefit from probenecid as an RSV prophylactic or treatment is likely very different. Could the authors discuss any existing data on its use in young children, or potential side effects from its use.
  • Probenecid has been tested in children 2 to 14 years of age for use together with antibiotics. It has not been shown to cause different side effects or problems than it does in adults. Studies on the effects of probenecid in patients with gout have been done only in adults. Gout is very rare in children. This has been added to the last paragraph in the revised Discussion.

  1. Please clarify whether the pharmacokinetics table (Table 2) pertains to humans, and if there are data to link mouse dosage to expected human effective dosage for probenecid.
  • We have clarified in Table 2 that the probenecid steady-state concentration and free drug concentrations shown are for different probenecid doses in humans. There is no data available to link probenecid treatment of mice to expected effective doses in humans.

Reviewer 2 Report

The paper from Murray et al analyze the antiviral properties of Probenecid as a valuable mean to fight respiratory syncytial virus. Their work, although purely pre-clinical, represent the first approach exploring this topic providing a worthy contribution to the existing literature. The manuscript is fairly written and its findings could provide basis for new research in the field. Specific comments follow below.

Introduction:

  1. Pg 1 lines 24-25. When stating “RSV is responsible 24 for >150,000 pediatric hospitalizations/year costing >$300 million in young children” the Authors should specify to which context does the data cited refer.
  2. When describing the anti-RSV therapies/vaccines the Authors should consider citing Nirsevimab as one of the most important novelties in the field potentially able to partially overcome some of Palivizumab’s weaknesses.
  3. Pg 1 lines 43-44 is not clear what the Authors mean by The antiviral drug approach…”. I think the paper could benefit by reformulating this sentence.
  4. Have ever preclinical evidence on the efficacy of Probenecid against “ influenza A virus” been confirmed by studies on human subjects?
  5. Pg 1 l. 85-86. In my opinion when referring to the cell lines “Vero E6” and “Hep-2” used it should facilitate the reader (especially if not used to laboratory practice) to briefly explain what kind of cell lines are those.

Though well written the “introduction” section could benefit from shortening and briefly specifying which has been the aim of the study in the final lines rather than describing the findings (these should be appropriately treated in the dedicated “results” section).

Materials and Methods

  1. 4 l. 167-169 in the section “statistical analysis” the article could be improved by briefly specifying the programmes used for statistics and graphs/figures production.

Discussion

This section should be substantially shortened. The real discussion of the results gathered seems to be too short.

  1. Pg 8 l. 269-291 The whole first part of the “discussion” section seems to repeat concepts already written in the “introduction”. The paper could substantially benefit from shortening that part as synthesis could ease reader’s comprehension.
  2. Why was probenecid prophylactic use better than its use as a therapy in “in vivo” model?
  3. Which could be the reasons for a better efficacy of probenecid against RSV A strain?

Author Response

Reviewer 2: Comments and Suggestions for Authors

The paper from Murray et al analyze the antiviral properties of Probenecid as a valuable mean to fight respiratory syncytial virus. Their work, although purely pre-clinical, represent the first approach exploring this topic providing a worthy contribution to the existing literature. The manuscript is fairly written and its findings could provide basis for new research in the field. Specific comments follow below.

Introduction:

Pg 1 lines 24-25. When stating “RSV is responsible 24 for >150,000 pediatric hospitalizations/year costing >$300 million in young children” the Authors should specify to which context does the data cited refer.

  • We agree and note that the information is based on health care costs of hospitalization of young children for respiratory syncytial virus infections, and replaced the earlier reference with an updated reference (PMID: 34667075).

When describing the anti-RSV therapies/vaccines the Authors should consider citing Nirsevimab as one of the most important novelties in the field potentially able to partially overcome some of Palivizumab’s weaknesses.

  • We clarify that the approved therapeutic intervention is limited to inhaled ribavirin and palivizumab (Synagis). Nirsevimab is in Phase 3 clinical trials and is not approved, therefore we chose to limit the discussion to approved drugs as there are many RSV drug candidates under investigation (PMID: 32096420), and it is not appropriate to focus on only one unapproved candidate. Although nirsevimab was shown to protect infants against RSV disease using one injection before the RSV season, in the same clinical study, nirsevimab treatment had no effect on hospitalization rates thus it is possible that it will have limited benefits for all infants.

Pg 1 lines 43-44 is not clear what the Authors mean by “The antiviral drug approach…”. I think the paper could benefit by reformulating this sentence.

  • We have clarified the sentence to read “Targeting the virus with antiviral drugs increases the likelihood of developing drug resistance”.

Have ever preclinical evidence on the efficacy of Probenecid against “ influenza A virus” been confirmed by studies on human subjects?

  • To our knowledge, there have been no probenecid studies on influenza virus replication reported in humans.

Pg 1 l. 85-86. In my opinion when referring to the cell lines “Vero E6” and “Hep-2” used it should facilitate the reader (especially if not used to laboratory practice) to briefly explain what kind of cell lines are those.

  • We concur and explain the cell types in section 2.1. Cells and Cell Culture.

Though well written the “introduction” section could benefit from shortening and briefly specifying which has been the aim of the study in the final lines rather than describing the findings (these should be appropriately treated in the dedicated “results” section).

  • We have shortened and modified the Introduction in the revised manuscript.

Materials and Methods

4 l. 167-169 in the section “statistical analysis” the article could be improved by briefly specifying the programmes used for statistics and graphs/figures production.

  • We now note in the revised manuscript that “Data were analyzed for statistical significance using appropriate statistics where p<0.05 was considered statistically significant using Prism 9 (GraphPad).”

Discussion

This section should be substantially shortened. The real discussion of the results gathered seems to be too short.

Pg 8 l. 269-291 The whole first part of the “discussion” section seems to repeat concepts already written in the “introduction”. The paper could substantially benefit from shortening that part as synthesis could ease reader’s comprehension.

  • We have shortened the Discussion in the revised manuscript.

Why was probenecid prophylactic use better than its use as a therapy in “in vivo” model?

  • It is likely that this difference reflects the PK/PD of probenecid in vivo and the rapid replication of RSV in the lungs. Probenecid would have adequate time to distribute to respiratory epithelial cells prior to infection.

Which could be the reasons for a better efficacy of probenecid against RSV A strain?

  • The IC50/IC90 of probenecid was marginally better against RSV A2 compared to RSV B1 but the effect was cell type-dependent. This may be linked to the source of cells, i.e. monkey (Vero) vs. Hep2 (human), and or their type, e.g. Hep-2 vs. NHBE cells. It is also likely linked to the cellular distribution and uptake of (PK/PD) of probenecid.

Round 2

Reviewer 2 Report

The paper is greatly improved through the suggestions of the reviewers.